# Effects of Low-Temperature Plasma-Sterilization on Mars Analog Soil Samples Mixed with *Deinococcus radiodurans*

**DOI:** 10.3390/life6020022

**Published:** 2016-05-26

**Authors:** Janosch Schirmack, Marcel Fiebrandt, Katharina Stapelmann, Dirk Schulze-Makuch

**Affiliations:** 1Astrobiology Research Group, Center for Astronomy and Astrophysics, Technical University Berlin (TUB), Berlin 10623, Germany; dirksm@astro.physik.tu-berlin.de or dirksm@wsu.edu; 2Biomedical Applications of Plasma Technology, Institute for Electrical Engineering and Plasma Technology (AEPT), Ruhr University Bochum (RUB), Bochum 44801, Germany; fiebrandt@aept.ruhr-uni-bochum.de (M.F.); stapelmann@aept.ruhr-uni-bochum.de (K.S.); 3School of the Environment, Washington State University, Pullman, WA 99164, USA

**Keywords:** plasma sterilization, *Deinococcus radiodurans*, Mars, regolith

## Abstract

We used Ar plasma-sterilization at a temperature below 80 °C to examine its effects on the viability of microorganisms when intermixed with tested soil. Due to a relatively low temperature, this method is not thought to affect the properties of a soil, particularly its organic component, to a significant degree. The method has previously been shown to work well on spacecraft parts. The selected microorganism for this test was *Deinococcus radiodurans* R1, which is known for its remarkable resistance to radiation effects. Our results showed a reduction in microbial counts after applying a low temperature plasma, but not to a degree suitable for a sterilization of the soil. Even an increase of the treatment duration from 1.5 to 45 min did not achieve satisfying results, but only resulted in in a mean cell reduction rate of 75% compared to the untreated control samples.

## 1. Introduction

In order to design a life detection mission to Mars, we were searching for an effective sterilization method as a negative control. At the same time the selected method should not affect soil properties, particularly not the organic content of a chosen soil. As a method of choice we tried low temperature Ar-plasma sterilization (below 80 °C), because heat sterilization like autoclaving might be too destructive to the organic contents of a soil. In a previous study by Stapelmann *et al.* [1] it could be shown that plasma sterilization was a suitable method for sterilization of certain parts of a space craft. In that study, stainless steel screws were spiked with spores of the two bacterial strains *Bacillus subtilis* 168 and *Bacillus pumilus* SAFR-032. The samples were exposed to increasing time steps of plasma (varying from 100 to 400 W, H_2_ or H_2_/O_2_ gas mixture [20 sccm]). The spore inactivation rates estimated via incubation and colony forming units (CFU) were below the detection level (10^−7^) when the spores were exposed to the plasma for at least 300 s.

The objective in our test series was to quantify survival rates of microorganisms mixed within Martian simulant soil by applying a low-temperature plasma. We used three different Martian regolith analogs and mixed the soil particles with cells of *Deinococcus radiodurans*, which has been shown to be especially radiation resistant [2,3]. Since the sterilizing effect of the plasma treatment is also based on UV photons [1], the radiation resistant *D. radiodurans* is a most suitable test organism.

## 2. Materials and Methods

### 2.1. Incubation Medium

*D. radiodurans* cells were incubated on 2× TGY medium, either on agar plates or in liquid. The 2× TGY had the following composition (per 1000 mL): 10 g Trypticase-Pepton BBL^TM^ (Becton, Dickinson and Company, Sparks, MD, USA), 6 g Bacto Yeast-Extract (Becton, Dickinson and Company) and 2 g d-Glucose-Monohydrate (AppliChem, Darmstadt, Germany). For the plates 15 g of agar (Carl Roth, Karlsruhe, Germany) were added to the mixture. The PBS buffer solution contained (per 1000 mL): 7 g Na_2_HPO_4_ × 2H_2_O, 3 g KH_2_PO_4_ and 4 g NaCl (all Sigma-Aldrich, Munich, Germany).

### 2.2. Growing and Handling of Cells

The cells of *Deinococcus radiodurans* R1T (=ATCC 13939T = DSM 20539T) were incubated on TGY plates for up to four days. A single colony was then transferred to 25 mL of liquid TGY and incubated until the cell density reached 3 × 10^7^ cells·mL^−1^ (counted microscopically with a Thoma chamber of 0.01 mm depth). The cell suspension was centrifuged (3000× *g*, 15 min), the supernatant was discarded and the pellet was washed with 25 mL of PBS buffer. The centrifugation was repeated and after discarding the supernatant the pellet was resolved in 2.5 mL PBS buffer.

The dissolved cells were transferred in PBS aliquots of 0.2 mL to sterile serum bottles with 1.25 g of Mars analog soils. The three Mars analog soils tested were JSC Mars-1A, P-MRA and S-MRA, which are described in detail in [4] (Table 1). The bottles were desiccated for 72 h inside a plastic box capped with a lid and filled with the drying agent Köstrolith^®^ (CWK, Chemiewerk Bad Köstritz GmbH, Bad Köstritz, Germany). After desiccation the dried samples were filled into 100 mL glass bottles (Figure 1a–d). One part was sent to the Ruhr University Bochum for applying plasma sterilization, the other part of the samples was used to estimate the starting cell numbers as a control (counts as described in Section 2.4.).

### 2.3. Plasma Sterilization Procedure

Two separate test runs were conducted with a double inductively coupled plasma system (DICP) operated at a pressure of 10 Pa. The first test consisted of all three regolith mixtures (JSC Mars-1A, S-MRA and P-MRA) and had an exposure time of 1.5 min. An argon-nitrogen mixture (100:5 sccm) was used with continuous power of 500 W, as it emits significantly higher amounts of radiation in the bactericidal wavelength region from 100 nm to 400 nm compared to an argon-oxygen mixture (100:5 sccm) [6]. The absolute values presented in [6] are not applicable to the results shown here, due to modification of the power coupling of the plasma system. Nevertheless, the significantly higher emissions of the argon-nitrogen mixture are shown. Absolute values of the radiation dose in the range from 100 nm to 400 nm of the system configuration used in this study were measured [7]. A possible radical that can interact with the samples is atomic nitrogen. As atomic nitrogen is not known for etching of organic materials, we assume that microbial inactivation in the argon-nitrogen mixture is based on radiation effects. Only JSC Mars-1A analog soil was used in the second trial. In this case, an argon-oxygen mixture (100:5 sccm) was used as it produces reactive oxygen species due to dissociation of molecular oxygen in the plasma. Since radiation-based sterilization is faster than sterilization due to oxidation, the treatment time was increased. As this leads to heating of the sample above 80 °C in the continuous plasma, we used a pulsed mode, switching the plasma on and off at a frequency of 1 kHz and a duty cycle of 10%. To account for this fact, the treatment time was increased to 45 min, yielding a plasma exposure time of 4.5 min. Furthermore, the power was increased to 1500 W, yielding a mean power of 150 W, as it is only applied 10% of the time. The plasma conditions are not matched to each other as the main objective was to observe to what degree the cells of *D. radiodurans* were affected by radiation or reactive species and not which component is more efficient. As the samples were placed in an ICP discharge, ions towards the samples are only accelerated by several tenth of eV. Thus, sputtering of the sample due to ion bombardment can be neglected as possible inactivation mechanism.

In parallel, control tests with only a vacuum treatment were done for the same time period. These controls were exposed to the vacuum inside of the plasma chamber for the same amount of time as the plasma samples, but not treated with the plasma. After the specific treatment the samples were sent back to the TU Berlin to determine the respective survival rates. An additional transport control sample was shipped to the Ruhr University of Bochum and sent back to the TU Berlin without any additional treatment, to estimate the cell loss during shipping of the samples. Details of the facility used in which the experiments were conducted are provided by [6].

### 2.4. Colony Forming Unit (CFU) Counts

After treatment the desiccated regolith samples were liquefied with 5 mL of PBS buffer. Each 0.1 mL of the resolved sample was plated on TGY agar plates in a serial dilution of 10^1^, 10^2^, 10^3^ and 10^4^ and incubated for two days at 37 °C. The growing colonies on the plates with the highest dilutions were counted to estimate the number of initial cells·g^−1^ for each regolith sample. A CFU test with desiccated cell regolith samples before shipment to Bochum served as a starting control of the cell numbers. All CFU counts were performed in triplicate.

## 3. Results

The *D. radiodurans* cell counts within all three regolith analogs and an exposure time of 1.5 min to the plasma revealed no significant reduction in cell numbers, except for JSC Mars-1A (Figure 2). A comparison of the cells count showed significant lower cell numbers before and after the plasma treatment, when compared to the transport control but not when compared to the vacuum control. There was a slight but significant reduction of 57% in cell counts in the JSC Mars-1A analog soil. For the other two tested regolith analogs (P-MRS and S-MRS) there was no reduction in cell numbers detectable.

In the second setup of the experiment, only JSC Mars-1A was tested because a significant reduction was shown in the first experimental setup. The exposure time in the second experiment was increased to 45 min to account for the pulsed mode of the plasma. Furthermore, changing the gas mixture from argon-nitrogen to argon-oxygen results in a plasma with many more of reactive oxygen species, but far less radiation in the range from 100 nm to 400 nm. The change to a radical oxygen species based plasma caused only a slightly increased reduction of cell numbers (Figure 3). The initial microbial count decreased from 2.7 ± 0.6 × 10^6^ cells·g^−1^ to 6.7 ± 2.3 × 10^5^ cells·g^−1^ after low-T plasma exposure, which equates to a mean survival rate of 0.25 (survival rate = N/N_0_; with N = cell number after plasma treatment and N_0_ = cell number of the starting control) and thus a total reduction rate of 75%. This compares to the reduction of 2.3 ± 0.5 × 10^6^ to 9.3 ± 2.3 × 10^5^ after exposing JSC Mars-1A samples with *D. radiodurans* cells in the first experiment only for 1.5 min, resulting in a mean survival rate of 0.43 (and thus a reduction rate of 57%). There is not much of an additional benefit to change the plasma from a radiation based to a reactive oxygen based approach. In both experimental setups there is no significant difference between the cell numbers of the vacuum control sample and the plasma sample, although both values are lower than the transport control.

## 4. Discussion

Our experiment to test the effect of low temperature (<80 °C) plasma sterilization on a Martian regolith analog mixed with cells of *D. radiodurans* revealed only a minimum inactivation of the cells in the samples. A slightly larger reduction of the cell numbers was possible, when argon-oxygen plasma was used, but insufficient to use the applied method as a negative control for life detection.

The sterilizing effect of the method is mainly based on chemically reactive species, ions, and (V)UV photons. Therefore, the observation of a high survival rate of cells of *D. radiodurans* is not that surprising, because this strain in known for its exceptional radiation resistance [3]. Nevertheless, the effectiveness of plasma for inactivating bacterial spores has previously been demonstrated for this setup [8], where a log 6 reduction of *Bacillus atrophaeus* was achieved within 60 s treatment time. With another low-pressure low-temperature plasma device, it was shown that it is possible to inactivate 10^8^ spores—contaminated dropwise on stainless steel screws—of the highly tolerant *Bacillus pumilus* SAFR-032 within 5 min treatment time [1]. Nevertheless, the experiments performed here, reveal only little effect of plasma on spiked Mars regolith analog. In addition to the high radiation resistance of *D. radiodurans,* the regolith particles might have helped the cells to survive the treatment. A thin layer of soil is capable to shielding the cells from harmful radiation, as could be shown previously [9]. A thin layer of an artificial meteorite powder helped spores to survive exposure of space conditions in the BIOPAN exposure experiment, whereas unprotected spores were almost completely inactivated. This is in good agreement of experiments done on spores by [10], who showed that even a few spore layers are sufficient to effectively protect the remaining spores against plasma. Although we used in our experiment only a very thin layer of soil (<1 mm) and *D. radiodurans* does not form spores, it seemingly was sufficient to help the cells to survive the treatment. Additionally, desiccated cells of *D. radiodurans*, as were used in this test, have shown an increased survival rate under various stress conditions [2]. The main effect reducing the cell numbers of *D. radiodurans* seems to be related to extreme dehydration rather than radiation and oxidative chemical species based on the similar results between plasma sterilization and vacuum control (Figure 3). This is also consistent with earlier experiments conducted by [11]. The reason for the different survival rates of *D. radiodurans* on the different regolith analogs in the first experimental setup is not fully understood. A possible explanation could be the different grain size distributions, which is most obvious for P-MRS *versus* JSC Mars-1A soil (Table 2 and Table 3). In P-MRS, the grain size component between 50 and 150 µm is dominant, which might have increased the shielding effect of the regolith against the influence of the plasma.

Thus, the method of low temperature plasma sterilization as it was used in our experiment was ineffective to reduce the cell numbers of a soil sample sufficiently. It appears this method is better suited for a cleanroom in spacecraft assembly facilities for treatment of surfaces, but based on our limited testing, not for sterilization if microbial cells are intermixed with soil.

## Figures and Tables

**Figure 1 life-06-00022-f001:**
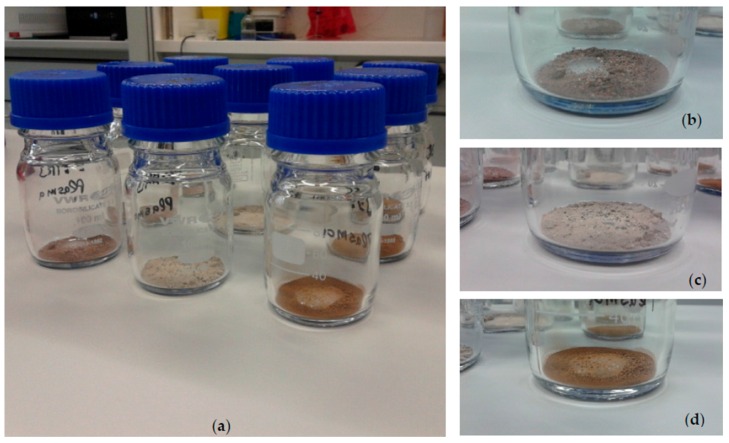
Desiccated regolith analog samples: (**a**) 1.25 g regolith mixed with cells of *D. radiodurans* in 100 mL glass bottles with wide opening and screw caps; Close ups showing (**b**) S-MRA; (**c**) P-MRA; (**d**) JSC Mars-1A.

**Figure 2 life-06-00022-f002:**
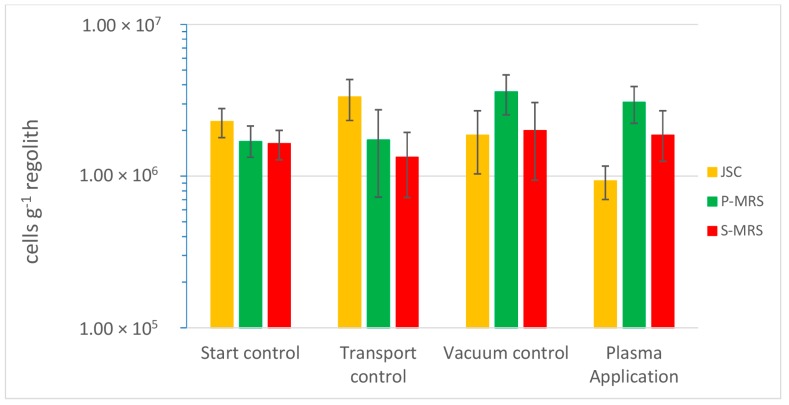
Results of the first experimental setup: estimated via plate count; Only for JSC Mars-1A a significant reduction in cell numbers could be observed before (Start) and after the treatment (Plasma). Error bars indicate the standard deviation, *n* = 3. The plasma sterilization parameters for this test were: Argon-Nitrogen Plasma (100:5) at *p* = 10 Pa, *P* = 500 W, *T* < 80 °C; duration: *t* = 1.5 min.

**Figure 3 life-06-00022-f003:**
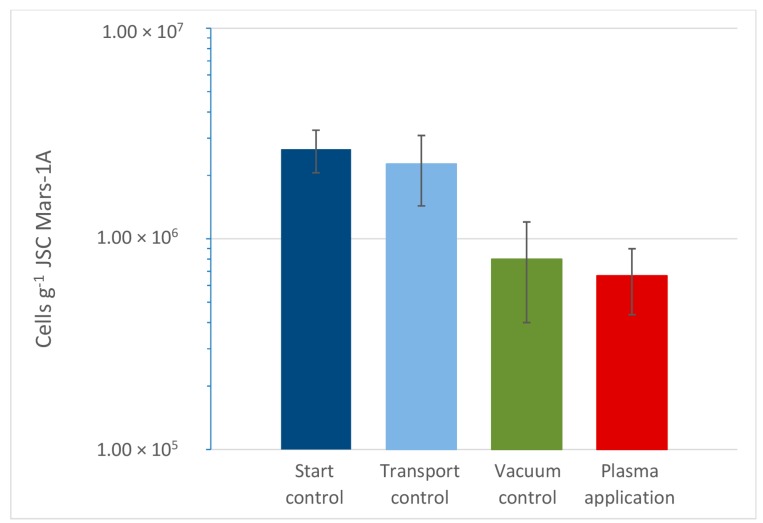
Results of the second experimental setup: estimated via plate count (here only JSC Mars-1A was used). Cell numbers of the plasma sample were reduced compared to the starting sample. No significant difference in counts between the plasma sample and the vacuum control sample could be observed. Error bars indicate standard deviation, *n* = 3. The plasma sterilization parameters of this test were: Argon-Oxygen Plasma (100:5) at *p* = 10 Pa, *P* = 1500 W (1 kHz, 10% duty cycle), *T* < 80 °C; duration: *t* = 45 min.

**Table 1 life-06-00022-t001:** Mineralogical composition of JSC Mars-1A, P-MRA and S-MRA: Composition as weight percent (wt %) of the mixture. Data for JSC Mars-1A were obtained from Morris *et al.*, 2010 [5]; data for P-MRA and S-MRA were obtained from Dr. Jörg Fritz, Museum für Naturkunde Berlin, Germany. (Adapted from Schirmack, *et al.*, 2015 [4]).

Mineral Phase	JSC Mars-1A (wt %)	P-MRA (wt %)	S-MRA (wt %)
Plagioclase Feldspar	64	-	-
Olivine	12	-	-
Magnetite	11	-	-
Pyroxene and/or Glass	9	-	-
Fe_2_O_3_	-	5	-
Montmorillonite	-	45	-
Chamosite	-	20	-
Kaolinite	-	5	-
Siderite	-	5	-
Hydromagnesite	-	5	-
Quartz	-	10	3
Gabbro	-	3	31
Dunite	-	2	16
Hematite	5	-	17
Goethite	-	-	3
Gypsum	-	-	30

**Table 2 life-06-00022-t002:** Grain size distribution of P-MRS and S-MRS: distribution as weight percent (wt %) of the mixture. Data were obtained from Jörg Fritz, Museum für Naturkunde Berlin, Germany.

Grain Size [µm]	P-MRS	S-MRS
1000–550	13%	30%
550–300	11%	26%
300–150	12%	16%
150–50	50%	14%
50–0	14%	14%

**Table 3 life-06-00022-t003:** Grain size distribution of JSC Mars-1A: distribution as weight percent (wt %) of the mixture*.* Data were obtained and adapted from Wan *et al.*, 2016 [12].

Grain Size [µm]	JSC Mars-1A
1000–500	20%
500–250	30%
250–150	20%
150–100	12%
100–50	10%
50–0	8%

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
