# Peer review of "Effects of Low-Temperature Plasma-Sterilization on Mars Analog Soil Samples Mixed with Deinococcus radiodurans"

_life, 2016, doi:10.3390/life6020022_

Reviewer 1 Report

In the manuscript ‘Testing the Effects of Low-Temperature Plasma-Sterilization on Mars Analog Soil Samples Intermixed with Deinococcus radiodurans’ by Schirmack et al., the results of the application of low-temperature plasma on Mars regolth simulant mixed with cells of the bacterium D. radiodurans were reported. Two different gas mixtures were used to produce plasma, argon-nitrogen and argon-oxygen. The exposure times varied from 1.5 to 45 min with different experiment Conditions/procedures without resulting in a significant inactivating effect on the bacterial cells.

Comments:

The plasma conditions are not described sufficiently. It is only stated that the argon-nitrogen mixture ‘mainly emits high amounts of radiation  in the wavelength region from 100 nm to 400 nm’ without specifying what high fluences at the sample site are. The term amount is not applicable here. With the knowledge of the spectral irradiance in the UV range from 100 to 400 nm the authors could have made a calculation to estimate the inactivation of D. radiodurans by the UV component of the plasma.  

The other potentially inactivating components of the plasma are not specified in the manuscript.  For the argon-oxygen gas mixture it is stated that ‘high amounts of reactive oxygen species are formed in the plasma’ again without any qualitative and quantitative description.  Estimations of the average free length of path for the inactivating plasma components might have been helpful for experiment planning.

Three different Mars regolith simulants were used. The mineralogical composition is given in table, but not the grain size distribution. Knowing the grain sizes and assuming a homogeneous distributrion of cells and grains or a measurement of the real distribution an assessment can be made about the shading effect of the grains with respect to UV radiation.

The authors should have estimated the maximal theoretically achievable effect and to plan the experiments accordingly.

Author Response

Reviewers comments:

The plasma conditions are not described sufficiently. It is only stated that the argon-nitrogen mixture ‘mainly emits high amounts of radiation in the wavelength region from 100 nm to 400 nm’ without specifying what high fluences at the sample site are. The term amount is not applicable here. With the knowledge of the spectral irradiance in the UV range from 100 to 400 nm the authors could have made a calculation to estimate the inactivation of D. radiodurans by the UV component of the plasma.

Reply: The authors would like to thank for this helpful comment. We changed the subsection as described below. With stating that argon-nitrogen mixtures mainly emit high amounts of radiation, the authors want to emphasize, that the sterilization efficiency is based on radiation rather than interaction of radicals with the sample. In low-pressure argon-nitrogen plasmas, the only significant possible radical is atomic nitrogen. As atomic nitrogen is not known for high etching efficiency towards organic materials, the main reason for inactivation of D. radiodurans is based on radiation effects. Furthermore, Argon-Nitrogen plasmas emit high amounts of radiation compared to Argon-Oxygen plasmas, as shown for example in [5]. The data in [5] is not comparable in absolute values, as the system was changed afterwards. Absolutely calibrated spectra were measured and have been submitted in another publication.

The authors think that estimating the inactivation efficiency towards D. radiodurans based on the UV dose cannot be performed on a reliable bases due to the following reasons: First, the inactivation efficiency towards spores is not only depending on the overall dose of the emitted radiation, but especially on the dose per wavelength. This has been demonstrated by Munakata et al. for B. subtilis spores (Munakata et al., Photochemistry and Photobiology, Blackwell Publishing Ltd, 1991, 54, 761-768). To the authors’ knowledge, there is no publication measuring the wavelength depending inactivation efficiency of B. radiodurans in the VUV. Second, the samples are mixed with Mars Analog Soil and most of the samples are covered with it. This leads to a very inhomogeneous exposure of the samples which can only be estimated arbitrarily.

To address the reviewer’s comments, subsection 2.3 has been modified as follows:

An argon-nitrogen mixture (100:5 sccm) was used with continuous power of 500 W, as it emits significantly higher amounts of radiation in the bactericidal wavelength region from 100 nm to 400 nm compared to an argon-oxygen mixture (100:5 sccm) [5]. The absolute values presented in [5] are not applicable to the results shown here, due to modification of the plasma system concerning the power coupling. Nevertheless, the significantly higher emission of the argon-nitrogen mixture is shown. Absolute values of the radiation dose in the range from 100 nm to 400 nm of the system configuration used in this study were measured and submitted [6]. A possible radical that can interact with the samples is atomic nitrogen. As atomic nitrogen is not known for etching of organic materials, we assume that inactivation in the argon-nitrogen mixture is based on radiation effects.

[6] Raguse, M.; Fiebrandt, M.; Denis, B.; Stapelmann, K.; Eichenberger, P.; Driks, A.; Eaton, P.; Awakowicz, P.; Moeller, R. Understanding of the importance of the spore coat structure and pigmentation in the Bacillus subtilis spore resistance to low pressure plasma sterilization. Submitted to Journal of Physics D: Applied Physics.

 The other potentially inactivating components of the plasma are not specified in the manuscript. For the argon-oxygen gas mixture it is stated that ‘high amounts of reactive oxygen species are formed in the plasma’ again without any qualitative and quantitative description. Estimations of the average free length of path for the inactivating plasma components might have been helpful for experiment planning.

Reply: The authors thank the referee for this comment that demonstrates that the authors need to clarify. However, to the authors it is not clear which potentially inactivating components of the plasma are not specified, as radicals and radiation are mentioned and the temperature is kept below 80°C to prevent inactivation effects due to heating. As the samples are placed in an ICP discharge, ions towards the samples are only accelerated by several tenth of eV. Thus, sputtering of the sample due to ion bombardment can be neglected.

Low-pressure plasmas dissociate molecular gases up to several percent. Thus, the authors wanted to emphasize, that the atomic oxygen density and therewith the atomic oxygen flux to the samples is in the order of a few percent of the overall gas flux. Quantitative and even qualitative analysis of atomic oxygen densities by optical emission spectroscopy is not an easy task as several parameters like gas temperature, electron density and electron temperature are necessary. This is even more challenging in pulsed mode, as the plasma does not achieve a steady state due to switching it on and off to prevent intensive heating of the plasma. Determination of the gas densities and plasma parameters is currently addressed in extensive analysis of plasmas with different gas compositions but not finished and published yet. Nevertheless, the authors’ state, that a relevant amount of atomic oxygen species is produced in the low pressure argon-oxygen plasma as the minimum energy for dissociation of molecular oxygen is around 5 eV. The ionization energy for argon is 16 eV and 13 eV for molecular oxygen. Thus, sufficient electrons with enough energy to dissociate molecular oxygen will be present in the plasma for dissociation up to a few percent.

In the authors’ view, estimating the mean free path of the atomic oxygen in the plasma is not necessary, as recombination of radicals in low pressure plasmas in the regime of a few Pascal takes place at the surfaces. Thus, any radical formed will be lost at the walls and not on its way to a surface. Estimating the mean free path of the radical in the Mars Analog Soil can only be performed on an arbitrary basis as the sticking and recombination coefficients of atomic oxygen are not known for the Mars Analog Soil. Thus, any calculation of the mean free path of the radical is based on estimated data and could only be verified by experiments.

To address the reviewer’s comments, subsection 2.3 has been modified as follows:

Only JSC Mars-1A analog soil was used in the second trial. In this case, an argon-oxygen mixture (100:5 sccm) was used as it produces reactive oxygen species due to dissociation of molecular oxygen in the plasma. Since radiation-based sterilization is faster than sterilization due to oxidation, the treatment time was increased. As this leads to heating of the sample above 80 °C in the continuous plasma, we used a pulsed mode, switching the plasma on and off at a frequency of 1 kHz and a duty cycle of 10 %. To account for this fact, the treatment time was increased to 45 min, yielding a plasma exposure time of 4.5 min. Furthermore, the power was increased to 1500 W, yielding a mean power of 150 W, as it is only applied 10 % of the time. The plasma conditions are not matched to each other as the main objective was to observe to what degree the cells of D. radiodurans were affected by radiation or reactive species and not which component is more efficient. As the samples are placed in an ICP discharge, ions towards the samples are only accelerated by several tenth of eV. Thus, sputtering of the sample due to ion bombardment can be neglected as possible inactivation mechanism.

Three different Mars regolith simulants were used. The mineralogical composition is given in table, but not the grain size distribution. Knowing the grain sizes and assuming a homogeneous distributrion of cells and grains or a measurement of the real distribution an assessment can be made about the shading effect of the grains with respect to UV radiation.

The authors should have estimated the maximal theoretically achievable effect and to plan the experiments accordingly.

Reply: The grain size distributions of the three Mars regolith simulants have been added to the text in form of two additional tables (table 2: P-MRS and S-MRS; and table 3: JSC Mars-1A), as well as a new citation for the grain size distribution of JSC Mars-1A [8].

To assess the shading effect of the regolith is difficult. As described we used a very thin layer of regolith particles, but did not measure them according to their shading effect but according to the weight, which was the same for all types of regolith. This experiment was meant as a test of the sterilization method adjusted to our purposes, as described in the introduction section. Therefore, we did not plan it according to the maximal theoretically achievable sterilization effect, but to design a life detection mission to Mars.

            [8] Wan, L., Wendner, R., & Cusatis, G. A Novel Material for In Situ Construction on Mars: Experiments and                 Numerical Simulations. 2016, arXiv preprint arXiv:1512.05461

Reviewer 2 Report

The manuscript contains scientific material that can be published as a short communication. However, there are some remarks on the justification of objectives of the study and presentation of the results.

First of all it is necessary to change the title and at least remove “The testing”. The introduction part requires more arguments to prove choice of the problem and the research object (D. radiodurans). In addition, I think, a control consisting of D. radiodurans cells before and after sterilization is missing to determine the role of the Martian Analog soils.

Minor comments

Lines 31-32: Bacillus subtilis and B.pumilus should be written in italic.

Line 38: Deinococcus radiodurans should be written in italic. While writing D. radiodurans R1 has always necessary to note that this is type strain R1T.

 Line 41: Medium instead of media. If I understood correctly, you used only one medium.

Line 50:  R1T (=ATCC 13939T=DSM 20539T) instead of R1 ATCC13939/DSM20539

Line 57: Are you sure that the Martian Analog soils were sterile before using?

Line 78: Please explain in the text  what is “a vacuum treatment control”.

Discussion

How do you explain the difference between the results when using JSC Mars-1A analog and the other soils?

Author Response

Reviwers comments

The manuscript contains scientific material that can be published as a short communication. However, there are some remarks on the justification of objectives of the study and presentation of the results.

First of all it is necessary to change the title and at least remove “The testing”.

Reply: The title has been changed.

The introduction part requires more arguments to prove choice of the problem and the research object (D. radiodurans).

Reply: In the introduction we already described the problem of the destructiveness of heat sterilization like autoclaving on the organic contents of a soil, which was our motivation to test an alternative method such as low temperature plasma sterilization. D. radiodurans was used as test organism because of its resistance towards UV radiation, which is a part of the sterilizing effect of plasma sterilization. The appropriate passages in the introduction section have been edited.

In addition, I think, a control consisting of D. radiodurans cells before and after sterilization is missing to determine the role of the Martian Analog soils.

Reply: Since we did not intend to test the influences of the Martian Analog soils on D. radiodurans, but the sterilizing effect of the low temperature plasma treatment on cells intermixed with soil, we did not make a control to determine the effects of the Mars analog soils. The starting values of cell numbers were estimated on samples prepared the same way as the samples for the plasma sterilization (intermixed with soils and desiccated as described in the method section). They were incubated on TYG agar plates like the plasma samples after the treatment. This explanation was missing in the text and has now been added to the appropriate passages in the method section (Chapter 2.2.) for a better understanding.

Minor comments (please see changes in text at the relevant lines)

Lines 31-32: Bacillus subtilis and B.pumilus should be written in italic.

Reply: The names are written in italics now, the mistake must have happened during the editing of the text.

Line 38: Deinococcus radiodurans should be written in italic. While writing D. radiodurans R1 has always necessary to note that this is type strain R1T.

Reply: Done, thank you.

Line 41: Medium instead of media. If I understood correctly, you used only one medium.

Reply: Yes, we just used one medium (TYG), so we changed it to ‘medium’.

Line 50: R1T (=ATCC 13939T=DSM 20539T) instead of R1 ATCC13939/DSM20539

Reply: The writing has been changed accordingly.

Line 57: Are you sure that the Martian Analog soils were sterile before using?

Reply: Since the bottles containing the Martian Analog soils were autoclaved before use, they should be sterile. We also made a separate test to check for sterility after autoclaving the Martian Analog soils, which has been negative (meaning no contamination could be detected via plating of the autoclaved JSC Mars-1A regolith on TYG medium).

Line 78: Please explain in the text what is “a vacuum treatment control”.

Reply: An explanation has been added to the text passage.

Discussion

How do you explain the difference between the results when using JSC Mars-1A analog and the other soils?

Reply: The differences might be caused due to the finer grain size of the other regolith used - particularly P-MRS, which possibly increased the shielding effect. This has been added to the discussion section.

The authors’ like to thank the reviewer for the helpful comments and suggested corrections to improve the manuscript.